# BIT CIPHER — A SIMPLE YET POWERFUL WORD REPRESENTATION SYSTEM THAT INTEGRATES EFFICIENTLY WITH LANGUAGE-MODELS

## ABSTRACT

While Large Language Models (LLMs) become ever more dominant, classic pre-trained word embeddings sustain their relevance through computational efficiency and nuanced linguistic interpretation. Drawing from recent studies demonstrating that the convergence of GloVe and word2vec optimizations *all* tend towards log-co-occurrence matrix variants, we construct a novel word representation system called ***Bit-cipher*** that eliminates the need of backpropagation while leveraging contextual information and hyper-efficient dimensionality reduction techniques based on unigram frequency, providing strong interpretability, alongside efficiency. We use the bit-cipher algorithm to train word vectors via a two-step process that critically relies on a hyperparameter—*bits*—that controls the vector dimension. While the first step trains the bit-cipher, the second utilizes it under two different aggregation modes—*summation* or *concatenation*—to produce contextually rich representations from word co-occurrences. We extend our investigation into bit-cipher's efficacy, performing probing experiments on part-of-speech (POS) tagging and named entity recognition (NER) to assess its competitiveness with classic embeddings like word2vec and GloVe. Additionally, we explore its applicability in LM training and fine-tuning. By replacing embedding layers with cipher embeddings, our experiments illustrate the notable efficiency of cipher in accelerating the training process and attaining better optima compared to conventional training paradigms. In fine-tuning experiments, training cipher embeddings on target datasets and replacing the embedding layer of the LMs to be fine-tuned negates the need for extensive model adjustments, offering a highly efficient transfer learning alternative. Experiments on the integration of bit-cipher embedding layers with Roberta, T5, and OPT, prior to or as a substitute for fine-tuning, showcase a promising enhancement to transfer learning, allowing rapid model convergence while preserving competitive performance.

## 1 INTRODUCTION

Word embedding algorithms serve as a crucial tools for understanding the semantics of categorical features in natural language processing (NLP) and deep learning (DL). Moreover, they continue to form an integral component of modern large language modeling (LLM) systems, since the initial step that LLMs, too, must approach is the efficient representation of tokens by static embeddings. Prior to the advent of the transformer architecture, it was research on pre-trained word embedding techniques that enabled DL for NLP. Pioneered by Mikolov et al. (2013a), word2vec ushered in NLP's era of representation learning, using the continuous bag-of-words and skip-gram models to demonstrate that it was possible to learn meaningful, low-dimensional representations with limited resources by predicting co-occurring tokens.

To accelerate learning via summary statistics (co-frequency), GloVe was ultimately introduced to harness the global statistics of co-occurrences (Pennington et al., 2014a), and moreover, without the use of contrastive learning. From there, it was ultimately a pivot to the modeling of sub-word information in a word2vec-like variant called FastText (Bojanowski et al., 2017b;a) that guided further pre-transformer advances, leaving us to ask:

> Could further improvements instead be made to the objective and optimization of embedding architectures, as opposed to their granularity of application?

Questions like this seem obscure with the arrival of the transformer, since the research paradigm has shifted from pre-trained word embeddings to more nuanced 'contextual' representations, defined by the hidden states of transformers. This shift saw the emergence of powerful models such as BERT, ELMo, and GPT (Devlin et al., 2019; Peters et al., 2018a; Radford et al., 2018; 2019), all of which relied on training LLMs to generate even higher-performance representations of words that demonstrate greater nuance at prediction of downstream tasks. However, since transformer embedding layers generally only leverage sub-word information (and positional encoding) over GloVe and word2vec, we see the presented main research question as not only valid, but by extension, capable of improving LLM architectures, since all require some form of embedding.

Notwithstanding the success of LLMs, one should still ask: Does the study of traditional word embedding methods retain any value? In this work, we argue in favor based on the following points: (1) the computational costs of training LLMs are substantial (Rae et al., 2022, Thoppilan et al., 2022), and obtaining contextual representations are essentially a by-product rather than the main objective of training LLMs. (2) Due to the cost-intensive nature of training LLMs, there is an inherent non-ideal trade-off between optimal performance and cost-effectiveness in them. (3) Initial pre-trained word embedding layers can greatly speed up and/or reduce the costs of training larger models that depend on embedding layers (Panahi et al., 2020).

In this work, we address points (1)–(3) by introducing the *bit-cipher*, which is a technique capable of representing words in a highly efficient manner into user-defined dimensionalities of word vectors. Drawing inspiration from one-hot encoding, the bit-cipher follows a straightforward and explicit process for vector assignment. Moreover, we extend capability by aligning with recent studies showing that the various forms of GloVe and word2vec converge towards variants of log-co-occurrence matrices. While we underscore the efficiency and competitiveness of bit-cipher against other pre-trained word embedding methods, we advise against using it in isolation or comparing it directly with contextual word embeddings. We perceive it more as a means to be used as a component of larger LM architectures, rather than as a standalone utility. In particular, we integrate contextual information via two different methods based on the summation (**Sum**) and concatenation (**Cat**) of co-occurrent information. Our investigations find that concatenation-based models using a large window size perform competitively when compared to GloVe and word2vec on Part-of-Speech (POS) tagging and Named Entity Recognition (NER) tasks, often out-performing both. Furthermore, experiments on integrating cipher embeddings into LM training and fine-tuning are conducted to show two of the main potential use scenarios of bit-cipher stating its efficiency and competitive nature with traditional methods.

## 2 RELATED WORK

**Generations and types of pre-trained word embeddings:** Representation learning in NLP has gone through many large transitions, starting from static word vectors (Mikolov et al., 2013a;b; Pennington et al., 2014b), and into contextual word representations (Howard & Ruder, 2018; Peters et al., 2018b), and now the predominant large language models (LLMs) (Devlin et al., 2019; Radford et al., 2018; 2019). These trends have often been based around architectural shifts, to/from the ubiquity of recurrent neural networks (RNNs) (Hochreiter & Schmidhuber, 1997), and then into reliance on attention mechanisms (Bahdanau et al., 2015) and the subsequent proliferation of self-attention leading to transformer-based LLMs (Vaswani et al., 2017).

**Optimization for pre-trained word embedding.** In the domain of pre-trained word embeddings, optimization methods are the essential that govern the performance and efficacy of the resulting word vectors. Early word embedding methods, like word2vec (Mikolov et al., 2013a) used gradient descent-based strategies to maximize context word likelihood, setting a foundation for subsequent models. Subsequent evolution led to the introduction of GloVe, which refined the optimization process by formulating a cost function based on global word co-occurrence statistics, merging local context and global matrix factorization methods to improve word representation. Despite its effectiveness, the performance of GloVe's optimization is limited by its predefined context window size to capture broader context (Pennington et al., 2014b).

Following significant development in the optimization of pre-trained word embeddings was revealed by Levy & Goldberg, 2014. They demonstrated that the skip-gram model with negative sampling implicitly executes matrix factorization on a word-context matrix which represents the pointwise mutual information (PMI) of the respective word-context pairs, emphasizing the critical role of matrix factorization in optimization techniques. This idea led to the understanding of how PMI-based word embeddings can encapsulate meaningful semantics (Arora et al., 2016). Recent study (Bojanowski et al., 2017a) further improved performance with subword embeddings, treating each word as a bag of character n-grams, particularly benefiting morphologically rich languages. Current research even extends these techniques to sentence and paragraph levels for more efficient representations (Arora et al., 2017).

**Dimensionality Reduction with Embedding.** Advances in dimensionality reduction have significantly contributed to word embeddings. Traditional techniques, such as PCA (Jolliffe, 1986) and SVD (Klema & Laub, 1980), transformed high-dimensional data into manageable lower-dimensional space, albeit with information loss. More recent works like Liu et al., 2016 introduced novel methods like Kernelized Matrix Factorization (KMF) that rejuvenated traditional matrix factorization techniques. Additionally, Heidenreich et al. (Heidenreich & Williams, 2022) elucidated the deep connection between word representation algorithms and co-occurrence matrix factorization. The BERT model (Devlin et al., 2019), despite its high-dimensionality, efficiently captures word semantics using dimensionality reduction techniques within a transformer architecture.

However, the techniques talked about above always involve training neural networks. A method that combines dimensionality reduction techniques with leveraging co-occurrence statistics for learning efficient word representations, without the need for neural network training to learn explicit representations of tokens, would be beneficial and ideal. As our primary focus, it will be discussed in the following section.

**Language Model training and fine-tuning.** In the days following the advent of the Transformer model, when Large Language Models (LLMs) were not as prevalent as they are today, the predominant method for utilizing Language Models (LMs) was through a process of pre-training and subsequent fine-tuning for specific downstream tasks. With the model size not as large as today's, it is not as expansive as fine-tuning a LLM. Consequently, 1) fine-tuning a language model for a specific purpose was less computationally intensive, and 2) the intrinsic properties of fine-tuning ensured that models could always achieve better performance through task-specific fine-tuning. However, with training language models at scale becoming possible and the dominant paradigm of carrying out NLP research, the cost of doing LM-related experiments has also increased. Despite the extraordinary power and utility of LLMs, the training process usually takes days and costs a huge amount of money while sometimes finding it hard to outperform smaller, fine-tuned language models (Liu et al., 2022) on specific downstream tasks. In section 5, we conduct experiments both for language model training and fine-tuning to demonstrate two useful scenarios to fit bit-cipher into the modern LLM world.

## 3 BIT-CIPHER

### 3.1 DEFINITION OF BIT-CIPHER

Standard-basis encoding is unavoidable for NLP applications, as one must always encode tokens from a given model's vocabulary: $W$. This makes dimensionality reduction *necessary* for NLP applications, as the combinatorial overhead on the model parameters required to process $|W|$-dimensional hidden states becomes tremendous inside of models. While dimensionality reduction can be handled via gradient-based optimization in DL systems, the random nature of DL optimization obfuscates the meaning of low dimensions. However, we conjecture that a similar and explicit encoder-decoder-style factorization of standard-basis information exists.

Supposing each token $t$ in $W$ has identity modified from the usual one-hot vector as follows: (1) select a 'low' dimension: $b \leq |W|$, and (2) assign a unique bit-vector, $\eta_t \in \{0,1\}^b$ to each. We base our approach on a distinguishability hypothesis: which expects that a 'good' order for the bits distinguishes the highest-frequency tokens best, and has latitude to assign similar-frequency tokens similar vectors, meaning word vectors are assigned based on unigram frequency ranking. Working along these lines, we define $b$-bit encipherment as the process of assigning probabilistically normalized (e.g., vectors are probabilistic vectors and the modulus of vectors is 1) with all $b$-bit

vectors in a 'smooth' order, by inducting the order that $i = 1$: assigns the set of $b$ standard basis vectors: $\mathcal{V}_1^b$ to the $b$ most-frequent tokens (generalizing one-hots/standard bases); $i = 2$: adds standard-basis vectors to those from $\mathcal{V}_{k-1}^b$ in reverse order of assignment, while filtering for unique bit-vectors in $\{0, 1\}^b$; $i = 3$: repeats step $i = 2$. $b$-bit vectors are then normalized for encipherment: $v_t = \eta_t / \|\eta_t\|_1$.

## 3.2 Modeling noise in observations

To assure that co-occurrence matrices are dense, we modify the base representation of the model from sparse, one-hot, to dense vectors of the same size. We first form a model: $\beta \in (0, 1)^N$, for the portion of time that each $i$-token's observations are (non-)erroneous as the definition shown above. Assuming that the highest-frequency tokens will be the least erroneously observed, we assume that only one error will be observed relative to each token's observed frequency, that is: $\beta_i = f_i / (f_i + 1)$, where $f_i$ is the unigram frequency of token $i$. Next and regardless of the token that is observed, we wish to modify its one-hot vector according to the probabilities that any different, $j$-token, should have been observed, instead, which will take the form of another vector: $\sigma \in (0, 1)^N$, but which is normalized: $\|\sigma\|_1 = 1$, and so define these other-token observation probabilities as: $\sigma_j = (1 - f_j / M) / (N - 1)$. To understand $\sigma$ intuitively, we first note that 1-minus each token's unigram probability: $1 - f_j / M$ expresses the probability of each token not being observed. Hence, the model $\sigma$ assumes that these (non-mutually exclusive) probabilities weight a distribution for the other token that should've been observed. For each one-hot vector, $y_i$, we then pull together these pieces to define noisy/dense vectors as: $\nu_i = \beta_i y_i + (1 - \beta_i)\sigma$, which form the embedding layers used in all language modeling architectures.

## 3.3 Rundown of procedurally building cipher embeddings

Knowing the definition and the encoding method of noise into cipher embedding enables us the procedural generation of word vectors. With the given dimension $d$, the bit-cipher algorithm is capable of generating the number of $2^d - 1$ vectors. In details of how the process works: The procedure operates in two steps: Initially, a set of probabilistic vectors, referred to as "plain vectors", is generated in accordance with the given *definition*. Subsequently, noise information is encoded based on the analysis of the ratio of document frequency to word frequency, denoted as $r_i = d_i / f_i$. This ratio determines the extent of noise information encoded into plain vectors. Specifically, words with high word frequency but low document frequency yield a small ratio, indicating that the word is noisy within the entire training set. Consequently, more noise information is "baked" into the plain vectors and vice versa. This is achieved using the formula: $\nu_i = \beta_i y_i + (1 - \beta_i)\sigma$ producing the final set of cipher embeddings The pseudocode of how exactly the algorithm can be implemented is also shown in Fig2

## 3.4 Illustration through a concrete sample case — 5-bit cipher

As depicted in Fig1, an example with 5 bits is illustrated. In this scenario, the bit-cipher algorithm can produce 31 distinct vectors with the capability of handling a corpus contain 31 unique tokens with each represented by a unique 5-bit vector. To elucidate the operation of the algorithm, consider the following steps visualized in the figure:

1. The first vector corresponds to the most frequent word in the corpus, assigning the bit-number 1 a value of 1, and all others a value of 0.

2. The second vector, representing the next frequent word, assigns bit-number 2 a value of 1, with all other bits set to 0. This pattern continues for the top $C(5, 1) = 5$ words, assigning a value of 1 to the corresponding position based

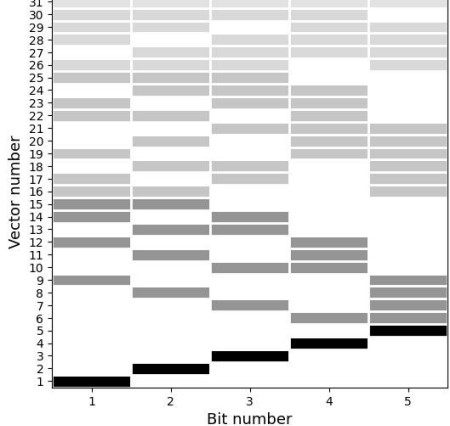

Figure 1: 5-bit example, carried out over its largest vocabulary size of $2^5 - 1 = 31$ vectors (rows).

1: **procedure** BIT-CIPHER$(N, b)$        ▷ Construct a $b$-bit cipher of $N \leq 2^{b-1}$ dimensions.
2:      $B^{(0)} \leftarrow [\vec{0}]$
3:      **for** $k = 1, \cdots, b$ **do**        ▷ **1.** Initialize sets for differently-normed bit-vectors.
4:          $B^{(k)} \leftarrow []$
5:     $U, V \leftarrow \{0\}^{N \times b}, \{0\}^{N \times b}$
6:     $i, j, k \leftarrow 0, 0, 1$
7:     **for** $n = 1, \cdots, N$ **do**
8:        **while** $V_n = \vec{0}$ **do**        ▷ **2.** Find the next norm-$k$ (or $k + 1$) bit-vector.
9:          $u \leftarrow \text{Abs}\left(B_j^{(k-1)} - I_i\right)$
10:          **if** $\|u\|_1 = k$ and $u \notin B_k$ **then**        ▷ **3.** The norm must be $k$ and the vector unused.
11:             $B^{(k)} \leftarrow \text{Concatenate}\left(B^{(k)}, [u]\right)$
12:             $V_n \leftarrow u / \|u\|_1$        ▷ **4.** Norm the bit-vector and assign it.
13:             $U_n \leftarrow u$
14:          $j \leftarrow j + 1$
15:          **if** $j = |B^{(k-1)}|$ **then**        ▷ **5.** Change basis vector/component of modification.
16:             $j \leftarrow 0$
17:             $i \leftarrow i + 1$
18:             **if** $i = b$ **then**        ▷ **6.** Reverse the $k$-bit vector order and increment $k$.
19:                 **if** $k = 1$ **then**
20:                    $I \leftarrow \text{Reverse}(I)$
21:                 $i \leftarrow 0$
22:                 $B^{(k)} \leftarrow \text{Reverse}\left(B^{(k)}\right)$
23:                 $k \leftarrow k + 1$
24:     **return** $U, V$        ▷ **7.** Return matrices for deciphering and enciphering.

Figure 2: Bit-Cipher algorithm. After 1) initialization, the algorithm must 2) find new bit-vectors in decreasing order of discernability, by 3) identifying bit-vectors of increasing norm (that have not yet been assigned) via translations of $k - 1$-bit vectors by standard basis vectors. Unassigned bit-vectors are then 4) normed for encipherment and assigned, along with the raw bit-vectors, which can be used for deciphering $b$-dimensional predictions. Whenever the collection of $k - 1$-bit vectors no longer has any unassigned $i$-component modifications, 5) the basis vector/component of modification must be incremented, and when this is the case for all last-component modifications, it's determined that there are no unassigned $k$-bit vectors, necessitating a 6) reversal of the $k$-bit vector order, which maintains smoothe transitions of discernability, upon future assignment. 7) Once all $N$ dimensions have been assigned a bit-vector (and normed counterpart), the matrices containing these vectors are returned.

on ranking.
3. Upon reaching the count of 5, words ranked between $[C(5, 1) + 1, C(5, 1) + C(5, 2)]$ e.g., $[6, 15]$ are assigned values in reverse order of index; two positions are assigned a value of $1/2 = 0.5$, and all others are 0.
4. For words ranked within the intervals of $[C(5, 2) + 1, C(5, 2) + C(5, 3)]$, $[C(5, 3) + 1, C(5, 3) + C(5, 4)]$, and $[C(5, 4) + 1, C(5, 4) + C(5, 5)]$, values are assigned to positions following the same logic.

Finally, each unique token is allocated a unique vector. By incorporating noise information relative to the distribution of words across various documents, the finalized version of bit-cipher embedding is obtained.

## 3.5 BIT-CIPHER TRAINING DETAIL

To illustrate the efficacy of cipher embeddings, models were trained on the CommonCrawl dataset using **Cat** (concatenation) and **Sum** (summation) methods for aggregating contextual information, informed by the *bits* hyperparameter, *log=True*, and *dtype='df'*. The latter two parameters enhanced sensitivity to infrequent words and adjusted noise levels based on word's document frequency, optimizing focus on distinctive words and mitigating biases.

The bit-cipher models are trained on five different scales of data-size: 0.5B-token, 1B-token, 2B-token, 4B-token, and 8B-token with setting up different radius (window size) and bits. Through incremental increase of the data-size, we aim to understand how model performance adjusts with the intake of more data. Although this data-size range is relatively small compared to other pre-trained word embedding methods, like GloVe trained with 42B and 840B tokens (Pennington et al., 2014a) and word2vec trained on Google News with 100B tokens (Mikolov et al., 2013a). All that is being said here is to validate the efficiency of bit-cipher as a means of learning representation, which we further corroborate through a series of probing experiments in the section.4.

Standard spaCy tokenization (Honnibal et al., 2020) was used for preprocessing, and models underwent a two-step training procedure as per sec 3.3 & Figure 2. Contextual information was integrated using **Cat** or **Sum** methods, with **Cat** models achieving representation lengths of 200d to 1600d (following the exponent of 2 times 100) across different bit settings and **Sum** models blending context information through element-wise addition, yielding a total of 60 models across varied radii and data sizes.

For comparability across models, we derived word embeddings from the GlOVe 6B embeddings, encompassing 400,000 tokens and with tokens appear in the evaluation datasets, yielding a total of 419,374 unique word embeddings. Any words identified within the context window that did not exist in our curated word-list were labeled as out-of-vocabulary (OOV) and were consequently assigned a distinctive embedding. This strategy for managing OOV words contributes to memory optimization, given that it mandates the processing of only a particular subset of words.

## 4 PROBING EXPERIMENTS FOR LINGUISTIC FEATURES CAPTURE

### 4.1 PROBING MODELS

The conduction of Probing experiments are inspired by (Hewitt & Liang, 2019) with designing POS tagging with the Georgetown University Multilayer (GUM) dataset (Zeldes, 2017), Named Entity Recognition (NER) using CoNLL-2003 shared benchmark dataset (Tjong Kim Sang & De Meulder, 2003) to evaluate the performance of bit-cipher.

**Named Entity Recognition.** NER probing experiment is conducted by CoNLL-2003 shared benchmark dataset which is a collection of data about Reuters newswire articles containing four different entity types: persons (PER), organizations (ORG), locations (LOC) and miscellaneous names (MISC). The probing model for NER is trained on CoNLL-2003 training data using CoNLL-2003 validation set for hyperparameter tuning. We follow the simplest and most straightforward setup with training an MLP by only using the bit-cipher embedding as the feature and directly adopt labels in the CoNLL-2003 dataset using the label-to-index method to convert each label into a unique number to setup the input and output of the probing model.

**Part-of-speech (POS) tagging.** Part-of-speech tagging is a task of assigning labels to each word with its corresponding grammatical category, such as noun, verb, adjective, etc. The Georgetown University Multilayer (GUM) dataset is a richly annotated corpus that contains comprehensive linguistic features. We extract the POS tagger of words in the GUM and train an MLP following the same setup as the NER experiment using the bit-cipher embeddings as the input and POS taggers as output.

### 4.2 PROBING MODEL BUILDING DETAILS

After obtaining the bit-cipher embeddings following 3.5, we applied a two-step post-processing to refine the word representations, and all probing experiments used this refined version of bit-cipher. Initially, a whitening transformation was employed to eliminate redundancies and normalize the embeddings, ensuring linearly uncorrelated word vectors with uniform variance, reducing inherent bias and making the distribution of embeddings more consistent (Kessy et al., 2016).

Next, we implemented mean-centering and L2 Normalization on each vector to address shifts in statistical distribution, inherent in probabilistic vectors like bit-cipher, which could cause inconsistencies in magnitude. This process stabilized the numerical representations, making them robust, and ensuring unbiased and scale-independent comparisons between word vectors.

For probing experiments, a 2-layer Multi-Layer Perception (MLP) was utilized, incorporating LeakReLU activation to mitigate the vanishing gradient problem, and a dropout rate of 0.5 for regularization (Xu et al., 2015). The output layer featured a LogSoftmax function, maintaining numerical stability and a balanced probability distribution, key for optimal performance.

### 4.3 PROBING EXPERIMENTS RESULTS

Probing experiments conducted on 100 separate bit-cipher embedding sets are presented in **Tables 2–7**. Their results at POS tagging and NER demonstrate noticeable and perhaps expected variations in performance. We see clearly that cipher-only models generally don't improve with increases of data (**Tabs. 6,7**), which is sensible given that ciphers only require ranking information and word frequency ranks converge over relatively little data.

For both **Sum** and **Cat** models (**Tabs. 2–4**), we see marked improvements from training over increased volumes of data, as observed from **Figure 3a & 3b** that when the *bits* is fixed, increasing the data size often results in improved model performance. Furthermore, in the case of Sum models, a clear performance gain is observed with an increase in the value of bits with bits = 200 set of models consistently demonstrate the highest performance. However, the performance is likewise sensible, assuming that the quadratic co-frequency information in co-occurrences requires more data to stabilize. However, *between* the **Sum** and **Cat** models, we note that **Cat** models improve over increases in data with greater *stability*—they scale more reliably as shown in **Figure 3a & 3b** that with fixing bits, 8B models always have the best performance. Moreover, we find that **Cat** models appear to consistently outperform same-dimension **Sum** models, de-

Table 1: Comparison of a 300-dimensional word2vec model against 200-dimensional models (all others) on probing experiments. Note: both of GloVe and word2vec were pre-trained externally using a larger radius of 10, by comparison to the Sum and Cat models presented, which were trained using $r = 4$.(values in the table are shown as accuracy with F1-score in Parentheses)

| Models | POS | NER |
|---|---|---|
| word2vec | 81.20 (80.80) | 78.55 (77.44) |
| GloVe.6B | 85.50 (86.09) | **91.70 (92.18)** |
| Cipher | 75.23 (73.58) | 86.19 (84.17) |
| Cipher (Sum) | 85.67 (86.04) | 90.67 (91.32) |
| Cipher (Cat) | **86.05 (86.32)** | 90.96 (91.51) |

spite being constrained to fewer bits, as can be seen from the cross-section of comparable models presented in **Tab. 1**. The inconsisency of model performance tendency is partially due to the fact that we did not do any preprocessing of the data except lowercase when training the bit-cipher. With refined preprocessing, the information gain would be even obviously to observe with the increase of data size.

When similar quantities of data are utilized, models that are more performant than word2vec, as well as quite comparable to GloVe, can be trained from bit-cipher co-occurrences. This can be see directly in **Tab. 1** for 200-dimensional bit-cipher models, which we compare to an externally-trained 300-dimensional word2vec model, a 200-dimensional GloVe and bit-cipher models. On its own, the noised cipher out-competes word2vec, while relatively-low radius ($r = 4$) **Sum** and **Cat** models perform comparably to a set of GloVe embedding, which were also externally trained. Despite the **Sum** and **Cat** models both utilizing a substantially smaller radius ($r = 4$) than GloVe ($r = 10$), we see that both of the comparable co-occurrent bit-cipher models out-perform GloVe at POS tagging, and perform comparably at NER. Finally, we note that these results *rank the bit-cipher at position 20* amongst the NER models retained on a well-known/public page: Tracking Progress in Natural Language Processing (Ruder, 2022), and moreover, present POS tagging results quite similar to other strong baselines (Ruder & Plank, 2018), whose model architectures tend to be much more complex and expressive than the MLPs used in our probing experiments.

## 5 EXPERIMENTS WITH LANGUAGE MODELS

Despite the models we've trained to exhibit the efficiency of cipher, and the probing experiments conducted to demonstrate its competitiveness with classic pre-trained word embeddings, we further explore what we believe to be two of the bit-cipher's most valuable applications — LM training and efficient LM fine-tuning.

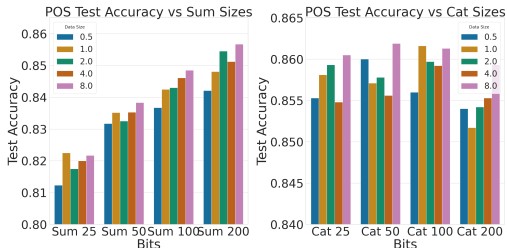
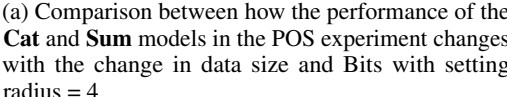
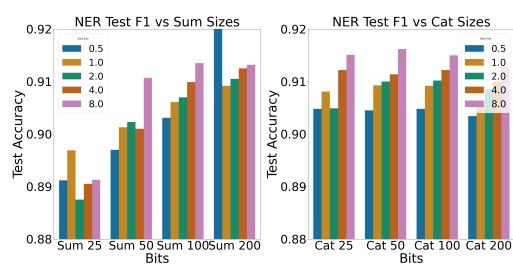

(a) Comparison between how the performance of the **Cat** and **Sum** models in the POS experiment changes with the change in data size and Bits with setting radius = 4

(b) Comparison between how the performance of the **Cat** and **Sum** models in the NER experiment changes with the change in data size and Bits with setting radius = 4

Figure 3: Comparison of **Cat** and **Sum** models in POS and NER experiments

## 5.1 A SHOWCASE OF BUILDING LANGUAGE MODELS WITH CIPHER

Firstly, the potential application of cipher is in the efficient training of language models. By using bit-cipher to construct the embedding layer of the language model and integrating it into the model's training process, we could potentially improve training efficiency and reduce the demand for computational resources.

Models are trained from scratch, utilizing both cold-start and warm-start approaches, with standard transformers (Vaswani et al., 2017). Our approach involves initially training bit-cipher with the BabyLM 10M dataset and replacing the randomly initialized embeddings in the warm-start model. An additional technique employed in the warm-start cipher with language model training involves freezing the embedding layer before the model is trained and subsequently unfreezing it for further optimization using backpropagation. This freezing/thawing technique offers two benefits: (1) As the embedding layer

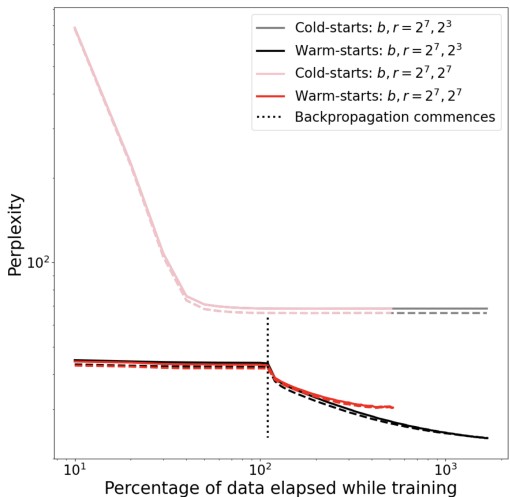

Figure 4: cold-start vs. warm-start perplexity curve with training processing

is the first layer in any language model, it typically requires the most optimization time through backpropagation and is thus the most expensive layer. By initially freezing this layer, (2) we avoid the deterioration of model performance, in terms of perplexity, that can occur when the sensitive and delicate embeddings are modified during warm-start training. Therefore, the warm-start model adopts a two-step training procedure: initially freezing the embedding layer and proceeding with regular training, followed by unfreezing the embedding layer for further optimization through backpropagation. Cold-start models adhere to the traditional training approach, initializing all parameters randomly and optimizing them through backpropagation.

We conducted experiments using two sets of cipher embeddings: one with bits=$2^7$ and radius=$2^7$, and another with bits=$2^7$ and radius=$2^3$. The comparison of perplexity between warm-start and cold-start models is illustrated in **Fig**4. The figure distinctly demonstrates that not only do warm-start models begin with a superior start, but they can also be further optimized through backpropagation, making this an overall more effective method for training language models.

## 5.2 LANGUAGE MODEL FINE-TUNING WITH BIT-CIPHER

In addition to using bit-cipher as part of LM training, we find it's also promising to use the algorithm for efficient LM fine-tuning. The traditional finetuning process, which necessitates retraining the model on a task-specific dataset, remains costly, leading to the exploration of zero-shot, few-shot,

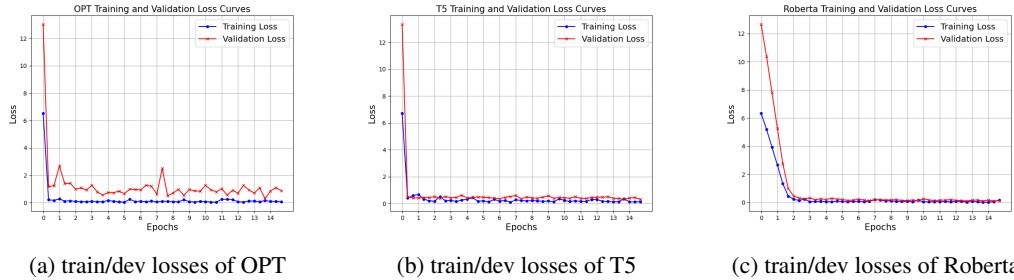

(a) train/dev losses of OPT    (b) train/dev losses of T5    (c) train/dev losses of Roberta

Figure 5: Loss curves of different models

and in-context learning strategies, prioritizing performance and efficiency. Although these methods effectively extract useful features learned during training, there is a known trade-off; for instance, prompted models do not always outperform fine-tuned models. Fine-tuned models, trained for a specific purpose on one or a series of related datasets for a downstream task, typically achieve state-of-the-art (SOTA) results.

A paradigm of fine-tuning that balances performance and training efficiency is desirable, allowing for the deployment of numerous specific-purpose models with superior performance that are less costly than training large models. In our method, we first train cipher embeddings on the fine-tuned dataset to acquire what we term "cipher fine-tune embeddings", then replace the embedding layer in the pre-trained language models with these cipher-fine-tuned embeddings, designed with specific fine-tune objectives. The efficiency of cipher training renders this step cost-effective, enhancing overall model efficiency.

We selected three language models: T5, Roberta, and OPT, and fine-tuned them on the 10M dataset provided by BabyLM (Warstadt et al., 2023). (Gao et al., 2021) are conducted following a two-step process: (1) Train cipher embeddings with the dataset used for the specific fine-tuning purpose. (2) Replace the embedding layer of the language model designated for fine-tuning with cipher embeddings. This approach enables models to converge more rapidly compared to traditional methods, as illustrated by three training/dev curves5, showcasing the speed of fine-tuning that fine-tuned model can quickly converge to low-enough training and developing losses which result in the acceleration of fine-tuning process as well as the reduction of computational costs.

## 6 CONCLUSION

In conclusion, in this paper, we introduce Bit-cipher, a novel and efficient method of learning word representations. By using this strategy, we acquire static pre-trained embeddings controlled by dimensionalities set with *bits* and learn contextual information by simple vector addition, eliminating the need for neural network training. Consequently, the model learns explicit statistical information from large text with strong interpretability. Our results show that **Cat** models consistently outperform **Sum** models across different dimensions and data sizes, demonstrating greater stability and superior performance even when constrained to fewer bits. However, **Sum** models also show merit, especially with the potential for further architectural improvements. Furthermore, by comparing with GloVe and word2vec, the competitive performance of bit-cipher is further validated. Additionally, language modeling experiments are conducted through both showing the efficacy of cipher as part of language model training and an efficient alternative to the traditional fine-tuning process. Overall, we see the bit-cipher as an efficient and high-performing alternative to classic pre-trained word embedding methods, with significantly reduced costs, offering a unique niche in the LLM era based on efficiency and interpretability—without performance compromise.

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

# A  APPENDIX

In appendix, we documented all the probing experiments results of all the bit-cipher models we trained both on POS tagging and NER with numbers in the tables are shown as accuracy with F1-scores shown in parentheses.

Table 2: Table for 60 **Sum** models of bit-cipher on POS tagging probing experiments

| bits | b = 25 | | | b = 50 | | | b = 100 | | | b = 200 | | |
|---|---|---|---|---|---|---|---|---|---|---|---|---|
| DataSize | r = 1 | r = 2 | r = 4 | r = 1 | r = 2 | r = 4 | r = 1 | r = 2 | r = 4 | r = 1 | r = 2 | r = 4 |
| 0.5B | 82.91 (83.08) | 82.70 (82.73) | 81.23 (80.85) | 84.35 (84.62) | 83.67 (84.02) | 83.17 (83.13) | 84.42 (84.52) | 84.67 (84.97) | 83.67 (83.91) | 84.89 (85.18) | 84.64 (84.87) | 84.21 (84.50) |
| 1.0B | 82.87 (83.04) | 82.36 (82.23) | **82.25** (**82.21**) | 84.52 (84.72) | 83.88 (83.95) | 83.52 (83.67) | **85.89** (**86.19**) | 84.97 (84.92) | 84.25 (84.32) | **85.43** (**85.65**) | 85.10 (85.40) | 84.81 (85.04) |
| 2.0B | 83.50 (83.78) | 82.91 (82.89) | 81.75 (81.40) | 84.55 (84.71) | 84.01 (84.18) | 83.25 (83.37) | 85.43 (85.70) | 84.92 (85.04) | 84.30 (84.47) | 85.42 (85.71) | 85.63 (85.92) | 85.45 (85.71) |
| 4.0B | 83.83 (83.87) | **83.07** (**83.08**) | 82.00 (81.87) | **84.85** (**85.01**) | 84.32 (84.50) | 83.53 (83.70) | 85.65 (85.90) | **85.64** (**85.90**) | 84.61 (85.00) | 85.18 (85.49) | 84.83 (84.87) | 85.12 (85.40) |
| 8.0B | **83.93** (**84.08**) | 82.85 (82.74) | 82.17 (81.96) | 84.44 (84.68) | **84.34** (**84.46**) | 83.83 (84.08) | 84.77 (84.99) | 85.28 (85.52) | **84.85** (**85.14**) | 85.36 (85.20) | **86.20** (**86.47**) | 85.67 (86.04) |

Table 3: Table for 60 **Sum** models of bit-cipher on NER tagging probing experiments

| bits | b = 25 | | | b = 50 | | | b = 100 | | | b = 200 | | |
|---|---|---|---|---|---|---|---|---|---|---|---|---|
| DataSize | r = 1 | r = 2 | r = 4 | r = 1 | r = 2 | r = 4 | r = 1 | r = 2 | r = 4 | r = 1 | r = 2 | r = 4 |
| 0.5B | **91.50** (**91.17**) | 89.08 (89.24) | 89.12 (89.12) | 89.60 (89.95) | 89.53 (89.80) | 89.12 (89.12) | 89.59 (90.02) | 89.78 (90.38) | 89.94 (90.31) | 89.47 (90.03) | 89.74 (90.33) | **92.35** (**92.35**) |
| 1.0B | 89.44 (89.75) | 89.26 (89.31) | **89.46** (**89.69**) | **92.23** (**91.94**) | 89.78 (90.11) | 89.84 (90.13) | 89.97 (90.41) | 90.22 (90.60) | 90.20 (90.61) | 89.95 (90.44) | 90.03 (90.56) | 90.30 (90.92) |
| 2.0B | 89.48 (89.74) | 89.58 (89.64) | 89.13 (88.75) | 90.04 (90.47) | 89.51 (89.55) | 89.93 (90.23) | 90.23 (90.81) | 90.38 (90.98) | 90.22 (90.70) | 90.20 (90.84) | 90.45 (90.91) | 90.50 (91.05) |
| 4.0B | 89.74 (90.06) | **89.74** (**90.01**) | 89.41 (89.05) | 90.25 (90.69) | **90.40** (**90.82**) | 89.77 (90.10) | 90.31 (90.81) | 90.38 (90.96) | 90.60 (90.99) | 90.31 (91.01) | 0.9042 (91.06) | 90.64 (91.25) |
| 8.0B | 89.97 (90.20) | 89.64 (89.56) | 89.32 (89.13) | 90.50 (91.03) | 90.31 (90.67) | **90.61** (**91.07**) | **90.65** (**91.24**) | **90.71** (**91.30**) | **90.82** (**91.35**) | **90.63** (**91.27**) | **90.72** (**91.25**) | 90.67 (91.32) |

Table 4: Table for 20 **Cat** models of bit-cipher on NER tagging probing experiments

| bits data-size | 25b (200d) | 50b (400d) | 100b (800d) | 200b (1600d) |
|---|---|---|---|---|
| 0.5B | 89.90 (90.48) | 89.93 (90.45) | 89.90 (90.48) | 89.83 (90.34) |
| 1.0B | 90.24 (90.81) | 90.49 (90.93) | 90.31 (90.92) | 90.18 (90.61) |
| 2.0B | 90.19 (90.49) | 90.42 (91.00) | 90.44 (91.02) | 90.28 (90.85) |
| 4.0B | 90.70 (91.22) | 90.74 (91.14) | 90.60 (91.22) | 90.49 (90.99) |
| 8.0B | **90.96 (91.51)** | **90.91 (91.62)** | **90.80 (91.50)** | **90.81 (91.25)** |

Table 5: Table for 20 **Cat** models of bit-cipher on POS tagging probing experiments

| bits
data-size | 25b (200d) | 50b (400d) | 100b (800d) | 200b (1600d) |
|---|---|---|---|---|
| 0.5B | 85.53 (85.88) | 86.00 (86.29) | 85.60 (85.81) | 85.40 (85.75) |
| 1.0B | 85.81 (86.35) | 85.71 (85.89) | 86.13 (86.44) | 85.17(85.57) |
| 2.0B | 85.93 (86.06) | 85.78 (86.24) | 85.97 (86.17) | 85.42 (85.88) |
| 4.0B | 85.48 (85.95) | 85.56 (85.74) | 85.92 (86.15) | 85.53 (86.04) |
| 8.0B | **86.05 (86.32)** | **86.19 (86.63)** | **86.16 (86.48)** | **85.93 (86.20)** |

Table 6: Table for 20 **Cip** models of cipher on its own on POS probing experiments

| bits
data-size | 25b | 50b | 100b | 200b |
|---|---|---|---|---|
| 0.5B | 73.76 (71.43) | 74.77 (73.08) | **75.31 (73.43)** | 75.21(73.56) |
| 1.0B | 73.65 (71.26) | 74.27 (72.44) | 74.64 (73.21) | **75.86 (73.92)** |
| 2.0B | **73.89 (71.50)** | 74.93 (72.94) | 75.49 (73.82) | 75.69 (73.90) |
| 4.0B | 72.21 (69.63) | 74.80 (73.06) | 74.93 (73.21) | 75.26 (73.68) |
| 8.0B | 72.72 (70.44) | **75.02 (73.41)** | 75.22 (73.46) | 75.23 (73.58) |

Table 7: Table for 20 **Cip** models of cipher on its own on NER probing experiments

| bits
data-size | 25b | 50b | 100b | 200b |
|---|---|---|---|---|
| 0.5B | 85.02 (83.55) | 85.64 (83.97) | 85.80 (83.92) | 85.83 (83.90) |
| 1.0B | 84.20 (82.72) | 85.58 (83.64) | 85.68 (83.61) | 85.75 (83.84) |
| 2.0B | 82.76 (82.20) | **85.75 (83.97)** | 85.75 (83.82) | 86.04 (84.14) |
| 4.0B | **85.22 (83.85)** | 85.66 (83.99) | 84.83 (83.50) | 85.98 (84.14) |
| 8.0B | 85.17 (83.83) | 85.54 (83.93) | **85.90 (84.03)** | **86.19 (84.17)** |

