# OpenReview forum: "Bit Cipher — A Simple yet Powerful Word Representation System that Integrates Efficiently with Language-Models"
_ICLR.cc/2024/Conference — ICLR 2024 Conference Withdrawn Submission_

### Official Review · Reviewer_eyQ8 · 2023-10-30

**Soundness:** 2 fair
**Presentation:** 2 fair
**Contribution:** 2 fair
**Rating:** 3
**Confidence:** 4

**Summary:**

This paper proposes a new method Bit-cipher to learn word representations efficiently while leveraging contextual information. It evaluation the efficacy of Bit-cipher on POS and NER tasks. The results of Bit-cipher is competitive to the two classic word embedding methods: word2vec and Glove. The paper also demonstrates the efficiency of Bit-cipher for LM training and fine-tuning with several   experiments intergrating LMs and Bit-cipher.

**Strengths:**

1. The idea of using bit-cipher to represent words is interesting.
2. Integration of word embeddings and LM is an important perspective to evaluate the proposed method.

**Weaknesses:**

1. The motivation of this paper is not strong enough. It lacks an explanation about the reason why bit-cipher achieves such performance. The advantanges of bit-cipher compared to the classic word embeddings are not clear. Only experimental results are provided. More theoretical proofs and straightfoward intuition should be provided.
2. The definition of bit-cipher is not clear. Section 3.1 is hard to follow. $\mathcal V_1^b$ is not introduced before using.
3. Probing experiments are conducted on only two downstream tasks (i.e., NER and POS). The results are only from one dataset for each task. The experiments are not convincing.
4. Again, to demonstrate the effeciency of Bit-cipher for LM integration, the experiments are not enough. For instance, it reach the claim "This approach enables models to converge more rapidly compared to traditional methods" without any comparison to classic word embeddings.

**Questions:**

1. What is the basic assumption of bit-cipher for learning word representations? What are the advantages of bit-cipher over classic methods?
2. Are there any theoretical proofs to support the advantages of the proposed method?

---

### Official Review · Reviewer_WMqp · 2023-10-30

**Soundness:** 2 fair
**Presentation:** 1 poor
**Contribution:** 2 fair
**Rating:** 3
**Confidence:** 3

**Summary:**

This paper proposes Bit-cipher, a word-embedding technique that eliminates the need of backpropagation while leveraging
contextual information and hyper-efficient dimensionality reduction techniques
based on unigram frequency, providing strong interpretability, alongside efficiency.
Experiments illustrate the notable efficiency of cipher in accelerating
the training process and attaining better optima compared to conventional training
paradigms.

**Strengths:**

* The proposed method is novel.
* This paper provides counter-intuitive results that a simple embedding algorithm could yield competitive performance.

**Weaknesses:**

* This paper is quite hard to follow, especially section 3.1 and 3.2. Many notations in those chapters are used without any clarification. I cannot figure out the method until I read section 3.4, which provides a concrete example of building cipher embeddings. Unless authors make great improvements on presentation in the next version, I lean to reject this work.
* In LLM finetuning experiments, I did not see comparation between cipher embeddings and Word2Vec or Glove.

**Questions:**

See Weakness

---

### Official Review · Reviewer_Py7Q · 2023-10-31

**Soundness:** 2 fair
**Presentation:** 2 fair
**Contribution:** 2 fair
**Rating:** 3
**Confidence:** 4

**Summary:**

The paper proposes a new word embedding approach called "bit-cipher". The authors test the approach in NER and POS, and use it for language models. The topic seems old in the large language model era, but this is fine. My main concern is that how the approach contributes to the community given that we already have Word2vec, Glove and many other variants; as a reviewer, I did not see a clear advantage of "bit-cipher" over existing methods.

**Strengths:**

- The topic is fundamental in NLP
-  The authors tried to use the proposed method in some modern language models like OPT and T5

**Weaknesses:**

- **sum** and **cat** seem vert common for NLP.
- The evaluated tasks like POS and NER, it may not aligned with the performance of downstream tasks.
-  The performance comparison should be compared with more careful way, such as aligning it with pre-trained corpora and parameter scales to have a apple-to-apple comparison.

**Questions:**

- What is the advantage of "bit-cipher" over Word2vec, Glove and many other variants?  what is the siginificance of "bit-cipher"? If it is insignificant,  people in the LLM era might not learn anything from it.  Please clarify the siginificance.
- For benchmarking, is it possible to consider some downstream tasks which are sensitive to word embeddings? The paper https://arxiv.org/pdf/1507.05523.pdf might help.